# Soil contamination in arid environments and assessment of remediation applying surface evaporation capacitor model; a case study from the Judean Desert, Israel

Rotem Golan[1], Ittai Gavrieli[2], Roee Katzir[1], Galit Sharabi[2], Uri Nachshon[1]

[1]Institute of soil, water and environmental sciences, Agricultural Research Organization, Volcani Center, Rishon-Letsioz, 7505101, Israel.
[2]Geological Survey of Israel, 32 Yisha'ayahu Leibowitz St. Jerusalem, 9692100, Israel.

*Correspondence to*: Uri Nachshon (urina@agri.gov.il)

**Abstract.** Due to the presence of highly pollutant industries in arid areas, many of the globe's arid areas are exposed to severe local soil contamination events. In this work, the nature of solutes and contaminants transport in sandy terraces of an ephemeral stream that was exposed to a severe pollution event was examined, taking the Ashalim basin in the Judean desert, Israel, as a case study.

In order to to shed new light on contaminants distribution along the soil profile and transport mechanisms, in arid environments, three complimentary approaches were used: (1) Periodic on-site soil profile sampling, recording the annual solute transport dynamics; (2) Laboratory analyses and controlled experiments in a rain simulator, to characterize solutes release and transport; and (3) Numerical simulation used to define and understand the main associated processes.

The study highlights the persistent nature of the pollutants in these natural settings, which dictates that they remain near the soil surface, despite the presence of sporadic rain events. It was shown that a vertical circulation of the contaminates is occurring with soil wetting and drying cycles. The 'surface evaporation capacitor' concept of Or and Lehmann (2019) was examined and compared to field measurements and numerical simulations, and found to be a useful tool for predicting the fate of the contaminants along the soil profile.

# 1 Introduction

Arid and semi-arid areas are defined as areas, beyond the polar and subpolar regions, where the ratio of annual precipitation to potential evapotranspiration is within the range of 0.05 to 0.65 (Gratzfeld, 2003). These areas, cover approximately 32% of Asia, 22% of Africa, 17% and 14% of North and South America, respectively, 7% of Europe, and 18% of Australia (Effat and Elbeih, 2020). Many of the earth deserts contain unique natural resources such as evaporative minerals (Reynolds et al., 2007), oil (Leaver, 1990), commercially important rocks such as phosphorites, and limestones (Abdel-Hakeem and El-Habaak, 2021; Sharma et al., 2000) and rare minerals or elements such as diamonds and uranium (Salom and Kivinen, 2020). Consequently, combined with low population density, low demand for agriculture lands and the low land cost, large areas in arid and semi-

arid environments are being used for industrial activities that involve highly polluting facilities. These include mines of different types (Abdel-Hakeem and El-Habaak, 2021; Effat and Elbeih, 2020; Portnov and Safriel, 2004), evaporation pans (Marazuela et al., 2020), oil production facilities (Luna et al., 2014), heavy and pollutant chemical industries (Dou et al., 2015; Effat and Elbeih, 2020; Portnov and Safriel, 2004) and landfills of different types (Cohen, 2007), including radioactive waste disposal sites (Shumway and Jackson, 2008).

Improper operational and management actions of these industrial activities, as well as uncontrolled and unpredicted accidents may cause severe soil, water and atmospheric contaminations. Often, such contamination events are associated with extreme climatic events such as floods (Sen et al., 2013; El Bastawesy and Abu El Ella, 2017; Becker et al., 2020; Gordon et al., 2018; Greenbaum, 2007; Izquierdo et al., 2020). With increasing risks for contamination events due to global climate change that includes higher probabilities of extreme rain events and floods (IPCC, 2007; Schewe et al., 2014), it is essential to better

understand the hydrological processes associated with such contaminations, in particular the nature of contaminants transport dynamics in arid environments and their fate thereafter. This understanding is a key for future prevention and mitigation of flood-associated contaminations in arid areas.

Natural flash flood events in arid environments occur during or following strong rain events. Water flow is typically restricted to the mainstream flow channel (MFC), whereas the elevated alluvial terraces are usually exposed to limited levels of moisture

(**Fig. 1A**). However, during rare and extreme rain events, which may also induce catastrophic pollution spills, the flow may reach areas beyond the MFC (**Fig. 1B**), exposing the rock slopes and terraces to unique hydrological conditions and processes. While a contaminated MFC is expected to experience natural processes of leaching, and contaminants removal during natural events of flash floods, the terraces and hill slopes are less likely to be exposed to significant natural wetting and flushing. Thus, the natural attenuation of the pollution in these terrains is expected to be significantly limited. Previous studies of water

infiltration processes and solute transport dynamics concentrated on the processes taking place beneath the MFC of ephemeral rivers in arid region, during and after flash floods (Amiaz et al., 2011; Basahi et al., 2018; Masoud et al., 2018). The present study focuses on governing solute transport processes that dictate the long-term fate of pollutants in the sandy terraces bordering the MFC. We also present a novel use of the surface evaporation capacitor (SEC) model of Or et al. (2019), as a tool to predict the fate of solutes in the examined system and to assess the likelihood of natural soil remediation, by soil leaching

with local rains. Our case study is a severe contamination event in the Ashalim Basin, an ephemeral stream in the Judean desert, described hereafter.

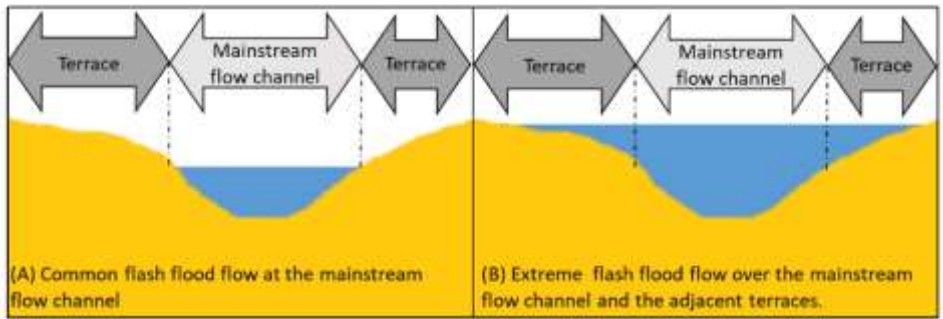

**Figure 1: conceptual representations of: (A) common flash flood event, where the flow is restricted to the mainstream flow channel; and (B) extreme flash flood, in which the uplands and terraces beyond the mainstream flow channel are exposed to the flood.**

### 1.1 The Ashalim Basin and contamination event of 2017

The Ashalim Basin, located in the Judean Desert, Israel (**Fig. 2A**), drains an area of ~80 square kilometers between Rotem Plain in the west and the Dead-Sea transform depression in the east (**Fig. 2A**). At its highest point in the west the basin is approximately 450 m above sea level, while its outlet elevation, ~15 km to the east is about 390 m below sea level, with the Ashalim stream course curving ~ 26 km (Becker et al., 2020; Greenbaum, 2007). In the upper, western part of the basin, the MFC curves through a 4-5 km sandy to carbonate terrain, and is characterized by a shallow and wide stream channel, with adjacent sandy terraces to the sides, bordering the hilly rock slopes (**Fig. 2B**). Towards the east, the landscape changes to a rocky terrain (mainly carbonates) and the stream channel narrows to form a deep gorge (**Fig. 2C**). The regional climate is arid with average annual precipitation of less than 100 mm, characterized by sporadic and short rain events during winter months, in between October and March (Zoccatelli et al., 2020). Thus, throughout most of the year the basin is dry whereas in winter flash floods may develop following strong rain events. Annual potential evaporation (PET) is higher than 2500 mm, as measured in a meteorological station located at the south-western edge of the basin, over a time span of nine years (2013-2022). The eastern part of the basin is part of the Judean Desert Nature Reserve (https://en.parks.org.il/map/).

The Rotem Plain complex, in the western part of the Ashalim basin, is a chemical-industrial zone (**Fig. 2A**). One of the major facilities on site is the 'Rotem-ICL' chemical plant, which produces phosphate from phosphorite rocks for the fertilizer industry and for other phosphate-based products. A by-product of the phosphate production is phosphogypsum, which is operationally stockpiled in high mounds overlooking the Ashalim basin. Operational ponds were constructed on the mounds' tops, to circulate the production fluids and deposit the phospogypsum. The latter accumulates at the bottom of the ponds, thus gradually elevating the acidic pond and forming the phosphogypsum stockpile. In June 2017, one of the dikes of the operational ponds breached, releasing about 300,000 cubic meters of highly acidic eluents and phosphogypsum slurry into the Ashalim stream (Becker et al., 2020; Rudnik, 2019). The acidic slurry contained also high concentrations of dissolved salts and heavy metals, derived from the phosphorite rocks as well as from the industrial process itself. The catastrophic spill event developed into a man-made flashflood in which flow levels reached elevation significantly higher than the MFC, covering large areas of the sandy terraces and bordering hillslopes, which commonly are not exposed to flow during natural flash-floods. Several years

85    after the event the mark of maximal height of the acidic flow is still easily observed in the field, in the form of a bright light
      coloring of the surface and sediments, in contrast to the above unaffected darker rocks and sediment (**Fig. 2B**). The pollution
      event caused an immediate extinction of local fauna and flora, including the death of an entire herd of the local desert mountain
      goat, the Nubian Ibex *(Capra nubiana).*

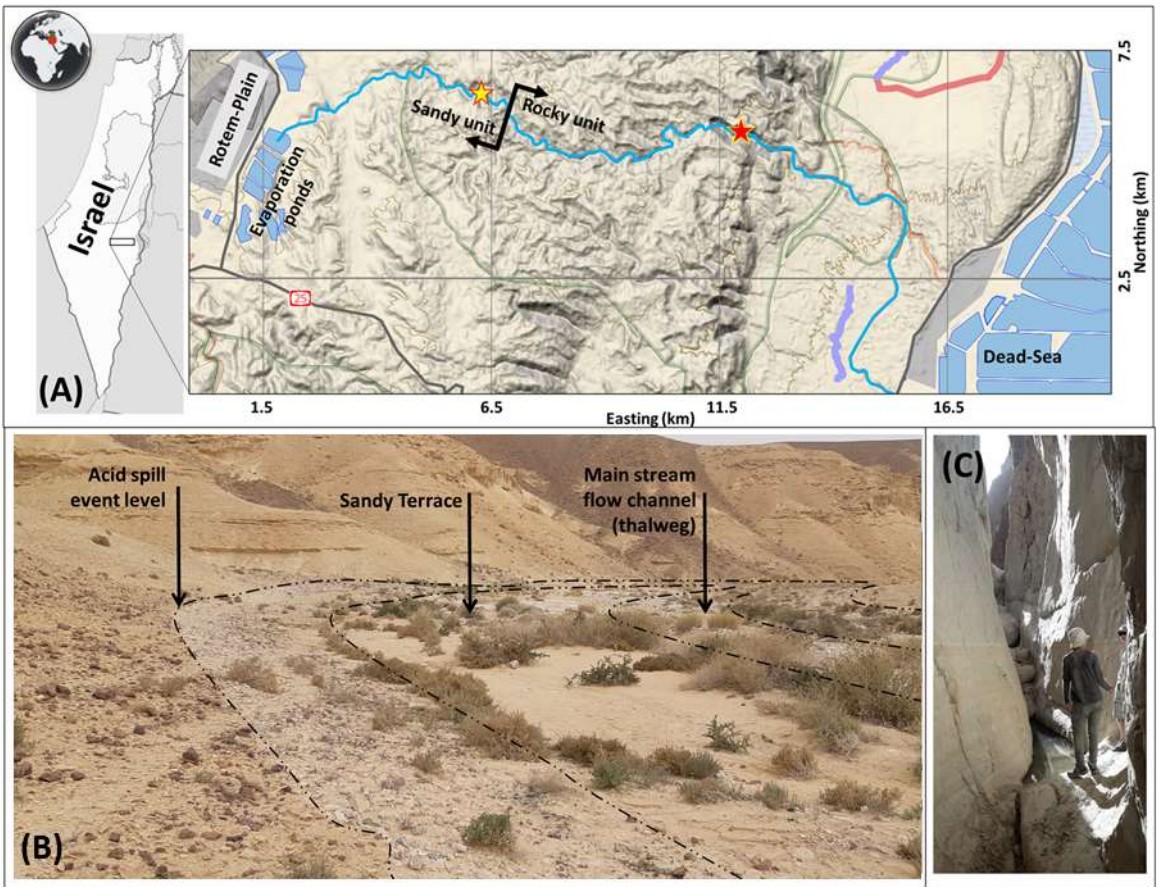

90    **Figure 2: Location map of the Ashalim Basin and the field site (A). The light blue curve follows the main channel of the
      Ashalim stream. Yellow and red stars designate the location of the sandy unit (picture B) and the deep gorge unit
      (picture C), respectively. Whitish coloring of surface in picture B is the coloring due to the spill event. The picture was
      taken at the vicinity of field monitoring station.**

      **1.2 Surface Evaporation Capacitor model**

95    Or and Lehmann (2019) termed the topsoil layer that contributes water to evaporation at soil surface during first stage of
      evaporation (S1) as 'Surface Evaporation Capacitor' (SEC). S1 functions as such, and evaporation is maximal, as long as there
      is hydraulic connection between the drying front and the soil surface (Lehmann et al., 2008). The SEC model predicts if pore
      water at the subsurface will be flowing by capillarity towards the evaporation front and evaporates, or if it will infiltrate

downwards to deeper soil levels and the groundwater, where the latter exists. Here, we use the SEC model to estimate the fate of conservative solutes and pollutants at the soil, which are being transported with the flowing pore water.

The thickness of the SEC layer that contributes water to evaporation at the soil surface is determined by the soil characteristic evaporation length, $L_C$ (m), which combines gravity and viscous forces, and is also determined by the potential evaporation at the soil surface, $E_0$ (m/s), and the soil hydraulic properties (Or and Lehmann, 2019):

$$L_c = \frac{L_G}{1 + \frac{E_0}{K_{eff}}} \tag{1}$$

where $K_{eff}$ (m/s) is the effective soil hydraulic conductivity within the SEC, and $L_G$ (m) is the maximum length for capillary flow against gravity to sustain evaporation from the surface. $L_G$ is determined solely by the soil physical properties, its texture and the difference in air entry pressure between the largest soil pores, $h_L$ (m), and the smallest hydraulically connected pore, $h_S$ (m).

$$L_G = h_S - h_L = \frac{(1-m)}{\alpha}\left(1 + \frac{1}{m}\right)^{(1+m)} \tag{2}$$

where

$$m = 1 - \frac{1}{n} \tag{3}$$

and $\alpha$ (1/m) and $n$ (-) are the van Genuchten parameters which describe the connection between matric suction and soil water content, for different soil textural properties (van Genuchten, 1983).

During S1, the effective soil hydraulic conductivity ($K_{eff}$) is affected by the matric suction of the small pores at the soil surface, and the soil hydraulic parameters:

$$K_{eff} = 4K_S\sqrt{(1 + (\alpha \cdot h_S)^n)^{-m}}\left(1 - \left(1 - \frac{1}{1+(\alpha \cdot h_S)^n}\right)^m\right)^2 \tag{4}$$

Where $K_s$ (m/s) is the soil hydraulic conductivity at saturation.

Determining the soil characteristic evaporation length, $L_c$, based upon soil properties and environmental conditions, enables to predict the flow direction of the pore water and solutes within, including contaminates, if present. In principle, following soil wetting event, a drop of water within the SEC may undergo one of the three following processes: (i) downward infiltration, to depths greater than $L_C$; (ii) upward flow to the soil surface, by capillarity, and evaporation at the soil surface; or (iii) no flow, provided that the SEC water content is low (~residual water content), and the hydraulic conductivity approaches zero, i.e., the pore water will not flow out of the pores. In this latter case, the water may be exposed to slow diffusive evaporation process, i.e., second stage of evaporation (S2). Downward Infiltration from the SEC initiates when the SEC mean water content exceeds a critical water content, $\theta_{crit}$, that corresponds to the matric suction of the smallest hydraulically connected pores of the medium $h_S$ (Or and Lehmann, 2019), and is equal to:

$$\theta_{crit} = \theta_{res} + (\theta_{sat} - \theta_{res})\left[1 + m^{m+1/m-1}\right] \tag{5}$$

For conservative solutes or pollutants that are being transported with pore water and do not interact with soil particles, the SEC model may be used to determine the fate and location of solutes accumulation. Water flow upward to the soil surface and evaporation during S1 will result in substantial accumulation of the solutes near the soil surface, and eventually may lead to salt precipitation above (efflorescence) or below (subflorescence) the soil surface (Dai et al., 2016; Nachshon et al., 2018). Water that infiltrates downward, to below the $L_C$ will carry away the solutes and the pollutants to deeper soil levels, removing them permanently from the soil surface and the root zone. Retained pore water, at conditions of residual water content will maintain the solutes in their location while S2 evaporation will concentrate them *in situ*.

Here we combine laboratory measurements, field observations, numerical modelling and estimation of the SEC thickness at the sandy soils of the Ashalim Basin to better understand pollutants dynamics and solute transport processes at sandy soils in arid environments.

## 2 Materials and methods

In order to better understand solute transport processes at the sandy terraces of the contaminated Ashalim Basin, two series of experiments were conducted in the laboratory; long term monitoring of solute distribution along the soil profile was conducted in the field; and a transport model was formulated, using HYDRUS-1D (Simunek et al., 2005). In addition, the SEC model was evaluated in the light of the field measurements and numerical model results.

The soil of the sandy terraces was characterized by the hydrometer method to determine soil particle size distribution (Ashworth et al., 2001), and the hanging column approach was used to define soil water retention properties (Schelle et al., 2013) and the van Genuchten parameters (van Genuchten, 1980). Hydraulic conductivity at saturation was measured by a Darcy test, over a 10 cm long tube, with pressure head difference of 25 cm. These hydrological properties, along with **Equations 1-4** and combined with field data of precipitation and evaporation, measured at a nearby meteorological station, were used to determine the thickness of the SEC layer at the sandy terraces of the Ashalim Basin. Field measurements and numerical model results were compared to and discussed in the light of the SEC model, as detailed below.

### 2.1 Laboratory experiments

Two laboratory experiments were conducted: (i) Batch leaching experiments, aimed to examine the dissolution and release dynamics of selected solutes out of the contaminated soil; and (ii) experiments in a rain simulator to study the release and transport of the solutes during rain events. For the experiments, contaminated sandy soil was collected from the contaminated terraces along the mainstream channel (yellow star in **Fig. 2A and B**). The soil samples were collected in June 2019, two years after the contamination event. Prior to any experiment or analysis in the laboratory, the collected soil was dried in an oven at 50°C for 48 hours, disintegrated by hand, and sieved through a 2 mm sieve.

### 2.1.1 Batch leaching experiments

In order to determine the composition of the major ions in the contaminated soils and the dynamics of dissolution and extraction of the substances from the soil, batch-dissolution experiments were conducted. 30 gr of the contaminated sandy soil and 30 ml of deionized water were mixed and shaken well for 10 minutes in 50 ml test tubes. The tubes were then centrifuged (3500 RPM) and the solutes decanted and filtered before being analyzed for their chemical composition, as detailed in **Table 1**. The solids in the test tube were then saturated again with 30 ml distilled water and the process repeated five times. The leaching experiments were done in triplicates.

**Table 1: Chemical analyses methods:**

| Parameter | Instrumentation | Error |
|---|---|---|
| EC, pH | Eutech PC 450, Thermo-Fisher, Waltham, MA, USA. | 0.01 dS/m; 0.02 pH units |
| Na, K | 420 Clinical Flame Photometer, Sherwood Scientific Ltd, Cambridge, UK | <10% |
| Ca, Mg | AAnalyst 400 Spectrometer, PerkinElmer Inc, Waltham, MA | <10% |
| Cl, $SO_4$, $PO_4$ | Gallery Discrete Analyzer, Thermo-Fisher, Waltham, MA, USA | <%5 |

### 2.1.2 Rain simulator experiments

The rain simulation experiments aimed to shed light on dissolution and transport processes of the solutes during rain events. The experiments were conducted in a Morin-type rainfall simulator, which enables to simulate rainstorms with different rainfall intensities on inclined surfaces (Morin et al., 1967). The simulated rain events were of a 43.0 mm cumulative rain, with rain intensity of 48.0 mm/h. The rainwater used for the simulations were deionized water, to closely mimic natural rainwater. For the experiments, the examined sandy soil samples were packed in tin trays, 4 cm deep, 45 cm long and 30 cm wide, which were placed in the simulator at a slope of 9%, with four replicates. Rain experiments were conducted for three setups: (i) the contaminated sand samples as collected from the field; (ii) the contaminated sand samples from the first rain experiment after being dried in the trays in free air for two weeks; and (iii) Control sand samples collected from a nearby stream which was not exposed to the contamination event, tributary of Gmalim stream, located ~1 km north of the Ashalim Basin. The control samples were not exposed to a consecutive rain event, as done for the contaminated samples, since it was assumed that they will be highly depleted of salts following the first rain event. This is presented in the results section below.

The rain simulator allows the collection of water infiltrating through the soil, as well as surface water. Infiltrating water samples were collected every few seconds at the beginning of each simulated rain event and every few minutes in the later stages of the experiments. Surface water, which had lower volumes compared to the infiltrating water were collected in ~20 minutes intervals. Water samples were analyzed in the laboratory for the chemical concentrations of selected components (**Table 1**).,

## 2.2 Field monitoring

Field measurements were conducted to characterize solute transport and accumulation processes at the terraces of the sandy
unit. On Sept. 29[th], 2020, two contaminated sandy terraces, located within the sandy unit (yellow star in **Fig. 2A; Fig. 3A**),
adjacent and above to the MFC were randomly selected to be monitored for their solutes transport processes over a one-year
period. The two sites were referred to as 'West plot' (31.077421°N35.256586°E) and 'East plot' (31.077722°N35.258020°E).
At each location soil profiling, down to a depth of ~50 cm, was done every few months. In addition, during winter, soil profiles
were collected following each rain event. For chemical analyses, the collected samples were oven dried for 48 hours at 50°C,
hand crushed, sieved, and mixed and shaken with deionized water at 1:1 ratio for two hours. Then chemical composition of
the extracted water was determined (**Table 1**). Mass water content ($\theta_m$) in the collected soil samples was determined   by
weighing the moist samples in a high-precision scale (Precision Balance PFB, Kern, Germany, 0.001 mg accuracy), prior to
and after oven-drying of the samples at 105°C for 24 hours. Then, the mass of water in the moist sample was divided by the
mass of the dry sample. Volumetric water content ($\theta_v$) was computed as:

$$\theta_v = \theta_m \frac{\rho_b}{\rho_w} \tag{6}$$

where $\rho_w$ is water density (=1 g/cm$^3$) and $\rho_b$ (g/cm$^3$) is the bulk density of the sample, which is proportional to soil porosity, $n$,
and the density of the solid particles in the sample, $\rho$, (commonly taken as 2.65 g/cm$^3$ ) (Skopp, 2001):

$$\rho_b = (1-n)\rho_s \tag{7}$$

$n$ was determined at the hanging column procedure, by weighing the amount of water needed to saturate the oven dry soil.
Daily rain depth data was available from the Israeli Nature and Parks Authority, that measured rain depth at a station located
~1 km northeast of the selected terraces.

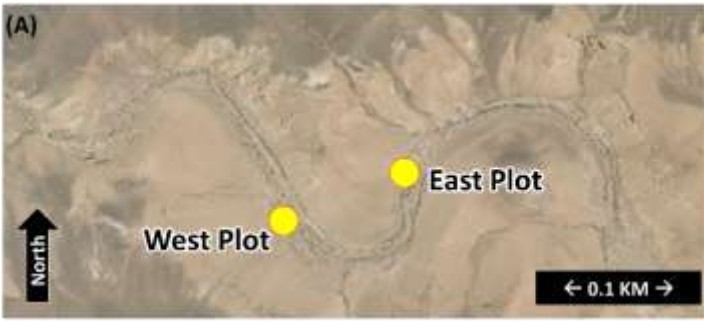

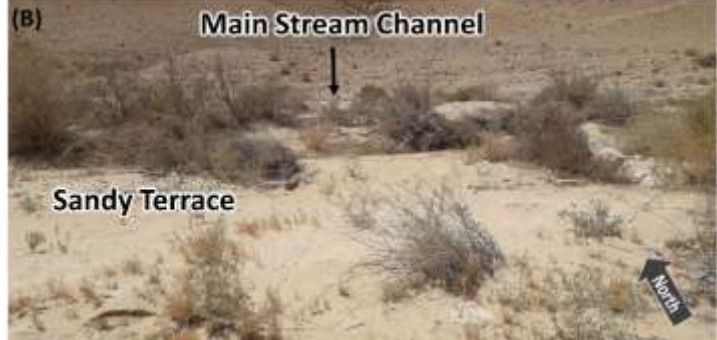

**Figure 3: Aerial location map view of the 'west' (31.077421ºN35.256586ºE) and 'east' (31.077722ºN35.258020ºE) monitoring plots (A); and characteristic view of a sandy terrace, specifically at the West Plot, adjacent to the mainstream channel (B).**

### 2.3 Numerical model

A numerical simulation in HYDRUS-1D (Schaap et al., 2001; Simunek et al., 2005) was applied to estimate the general leaching potential of solutes in sandy soils under arid climatic conditions. For this purpose, a one dimensional model was constructed and solute transport was simulated over a one-year course. The modelled domain was composed of a sandy soil with hydrological parameters as determined for the collected soil samples by the hanging column and hydrometer procedures (Ashworth et al., 2001; Schelle et al., 2013). Initial conditions were set as matric suction of -100 cm throughout the entire domain, corresponding to saturation degree of residual water content. The simulated polluted layer was represented as a 10 cm top layer containing a conservative solute. The solute concentration was set with a linear distribution of 10 and 0 mmol/cm$^3$ at the soil surface and the lower boundary of the 10 cm layer, respectively. Domain's lower boundary, at depth of 100 cm, was set as free drainage, and upper boundary was set as atmospheric boundary, with precipitation events corresponding to daily rain depths as measured at the Ashalim Basin during the winter of 2020-2021. Daily potential evaporation was set following long-term measurements conducted in the nearby meteorological station (**Table 2**). In addition, in order to study and better understand the impact of rain depth on solutes removal, similar simulations were performed with rain depths two and five fold higher than measured rains. The excess rain simulations were added to test and emphasize the effect of different rain depths on solutes leaching processes in high aridity locations. Note the scarce daily precipitation in the Ashalim region alongside the calculated aridity index (precipitation/potential evaporation), indicating high levels of aridity for the region (**Table 2**).

**Table 2: Averaged daily potential evaporation and precipitation over a year (data from a nearby meteorological station of the Israeli Nature and Parks Authority 2013-2022). The aridity index is the ratio between precipitation to potential evaporation.**

| Month | 1 | 2 | 3 | 4 | 5 | 6 | 7 | 8 | 9 | 10 | 11 | 12 |
|---|---|---|---|---|---|---|---|---|---|---|---|---|
| Averaged daily potential evaporation, (mm) | 3.2 | 4.2 | 5.6 | 8.4 | 10.1 | 11.0 | 10.9 | 9.6 | 8.1 | 6.6 | 4.9 | 3.3 |
| Averaged daily precipitation (mm) | 1.2 | 1.1 | 0.5 | 0.5 | 0.0 | 0.0 | 0.0 | 0.0 | 0.1 | 0.3 | 0.5 | 0.6 |
| Aridity index (-) | 0.38 | 0.26 | 0.08 | 0.05 | 0.00 | 0.00 | 0.00 | 0.00 | 0.01 | 0.04 | 0.10 | 0.20 |


## 3 Results and discussion

The results will be presented in the following order to allow meaningful discussion; First, the physical measurements and hydrological properties of the examined sandy soil at the Ashalim Basin will be introduced. Following, the dynamic release of solutes from the soil will be discussed based upon the batch extraction experiments and the rain simulator experimental results.

Next, the long-term field monitoring measurements will be presented, followed by the numerical model results alongside with the SEC model.

### 3.1 Soil physical properties

As detailed above, all laboratory experiments and analyses were done for soil samples collected in the field from the sandy terraces in the Ashalim Basin. Soil texture as defined by the hydrometer method and the USDA soil texture triangle (Shirazi

and Boersma, 1984) was characterized as sandy loam, corresponding to 79% sand, 1.5% silt, and 19.5% clay. **Table 3** presents hydrological properties of the soil, as measured by the Darcy test and hanging column method.

**Table 3: Hydraulic properties of the examined sandy loam soil.**

| Property | Value |
|---|---|
| Residual water content, $\theta_r$ [-] | 0.02 |
| Saturation water content, $\theta_s$ [-] | 0.37 |
| van Genuchten $\alpha$ [1/cm] | 0.03 |
| van Genuchten $n$ [-] | 2.5 |
| Saturated hydraulic conductivity, $K_s$ [cm/d] | 250 |
| Soil bulk density, $\rho_b$ [g.cm$^3$] | 1.67 |

### 3.2 Solutes release

As detailed above, solutes release was examined experimentally by the batch and rain simulator experiments. Both tests pointed

on similar dynamics of salts and minerals dissolution and solutes release, as detailed below.

### 3.2.1 Batch experiments

The dynamic extraction of solutes from the soil to the water, as measured in the five cycles of wetting and extraction of the batch experiment, is presented in **Figure. 4**. Over the five cycles of leaching EC dropped from average value of ~2.9 dS/m to ~2.2 dS/m (**Fig. 4A**). Concurrently, Na, Cl, K, and Mg concentrations were largely depleted by the five cycles of extraction, with most of the depletion occurring during the first two cycles. The observed rapid extraction of these solutes from the polluted soil indicates that they most likely originated from pollution-associated salts with relatively high solubility, such as NaCl, $Na_2SO_4$, KCl, and $MgSO_4$. In addition, their relatively low initial concentrations are indicative of lower quantities of these salts in the soil itself. In contrast, Ca, and $SO_4$ maintained relatively constant concentration levels over the five cycles of extraction, with close to 1:1 stoichiometric relation, pointing on gypsum ($CaSO_4 \cdot 2H_2O$) as the likely source mineral, in line with one of the main components in the Ashalim pollution event (phosphogypsum). All extracts are saturated with respect to gypsum (SI = 0.02, calculated by Phreeqc Interactive 3.7.3, using phreeqc database  (Parkhurst and Appelo, 2013)). Noteworthy is that their dominance in the solution determines the overall salinity and the EC in the last leachings.

A slight decline in $SO_4$ concentration is seen between cycles #1 and #2 (from ~33 to ~30 meq/l), while Ca concentrations remain relatively stable, at ~34 meq/l. This decline could point to the presence of minor amounts of relatively highly soluble sulfate minerals such as sodium sulfate and magnesium sulfate.  The slight excess of Ca over $SO_4$ in the remaining cycles is likely due to dissolution of some carbonate minerals, which are present in the local sands.

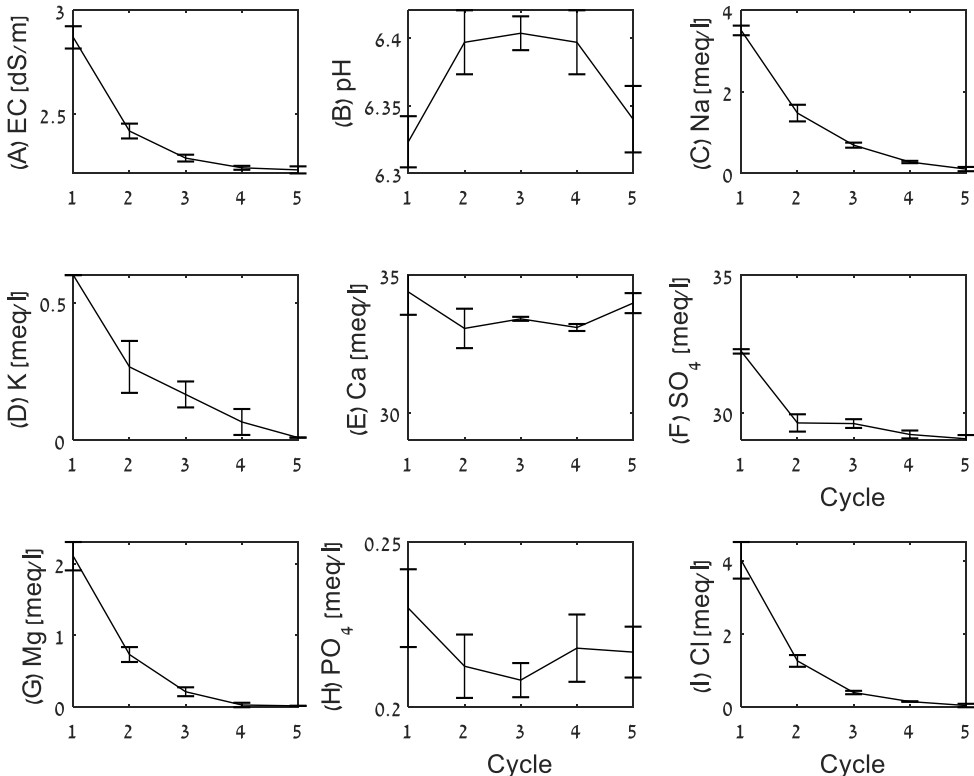

**Figure 4: Chemical properties of leachate water from the five cycles of batch experiments. Error bars represent standard deviation of the three triplicates.**


### 3.2.2 Rain simulator experiments

In the rain simulator experiments, more than 95% of rainwater percolated through the soil, and less than five percent of the rainwater drained along the inclined sandy slopes as runoff (**Fig. 5**). This means that for the examined conditions, the potential of lateral transport of salts and other contaminates by surface water is limited as most of the rainwater infiltrates downward
through the soil.

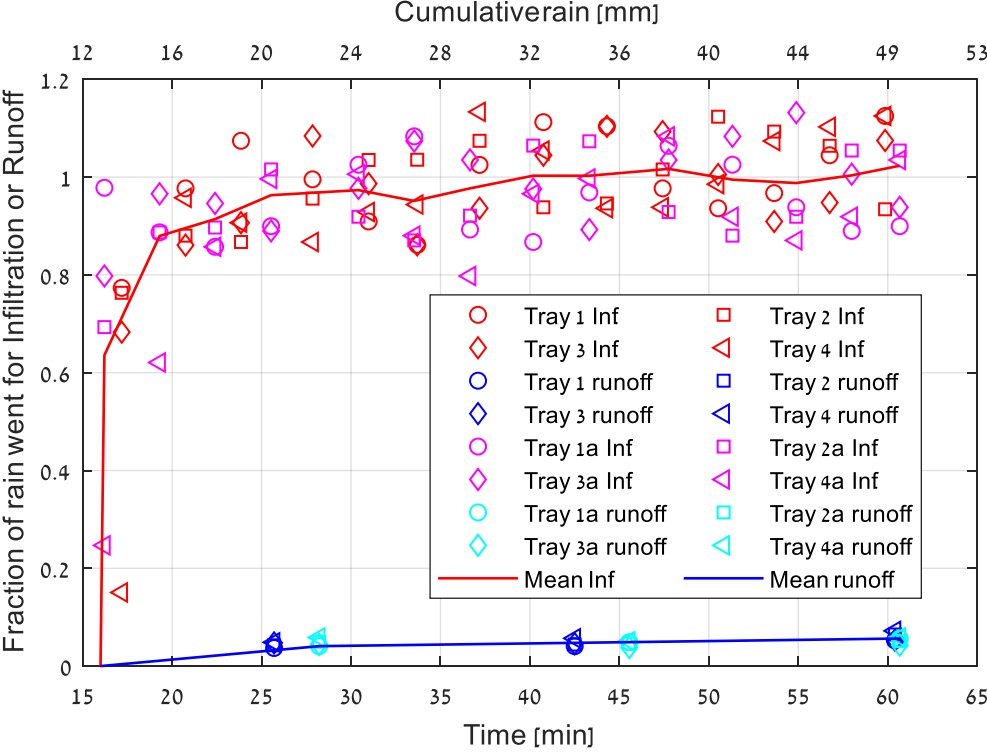

**Figure 5: Fractions of rainwater volume infiltrated through the soil layers (red and pink symbols) and surface runoff water (blue and aqua), as measured in the rain simulator experiments. All measurements of the four repetitions (Trays) and the two simulated rain events are marked as detailed by the legend.**


The patterns of solutes release observed in the batch experiments were also found in the rain simulator experiments, in which the concentrations of Na, Cl, Ca, and $SO_4$ were measured in samples from surface runoff and water that infiltrated through the soil (**Fig. 6**). The initial Na and Cl concentrations in the infiltrating water samples were at the order of ~40 meq/l, with the concentrations quickly dropping off. The most likely source for these solutes is the highly soluble halite (NaCl). These initial

concentrations are much higher than those measured in the batch experiments since the water to soil ratio in the batch experiments (1:1) was higher than the ratio of the first infiltrating water collected from the simulated rain events. The water to soil ratio of the infiltrating water in the rain simulator experiment was estimated by dividing the cumulative mass of collected water at the first reading of Na and Cl concentrations (15 mm rain depth), by the mass of soil through which the water percolated. The mass of soil was computed for a 4 cm soil thickness, with estimated density of 1.7 g/cm$^3$, yielding a water to

soil ratio of only ~1:7, which explains the elevated Na and Cl concentrations in this first infiltrating water sample.

In agreement with the batch extraction results, Na and Cl concentrations in the infiltrating water rapidly declined during the simulated rain event and after a cumulative rain of ~30 mm, both ions concentrations were reduced to levels of about 2 meq/l, similar to the control (**Fig. 6A and 6B**). These concentrations drop indicate a fast and efficient dissolution process and transport

of the dissolved Na and Cl ions downward with the infiltrating water. In line with that, in the consecutive simulated rain event, the initial concentrations of Na and Cl were relatively low, on the order of 5-10 meq/l, indicating that the reservoir of this salt in the washed soil layer was effectively lowered compared to the initial conditions of the first simulation. However, the Na and Cl concentrations at the beginning of the second rain simulation (using the contaminated sand from the first rain experiment after being dried in the trays in free air for two weeks) are higher than the concentrations measured at the end of the first rain event. This is likely related to accumulation of salts near the soil surface during the drying and evaporation period between the two simulated rain events, a result of capillary water flow towards the evaporation front at the soil surface. This process enforced the accumulation and precipitation of NaCl at the soil surface. Consequently, the water infiltrated through the soil at the beginning of the second rain event dissolved the cumulated salt and pushed the solutes downward, at relatively high concentration. Similarly to the first rain event, after cumulative rain of ~20 mm, both Na and Cl concentrations in both the infiltrated and runoff waters were reduced to low levels, of less than 2 meq/l.

Contrary to the above, Ca and $SO_4$, which are associated with the phosphogypsum that was deposited during the pollution event, and is much less soluble than halite, maintained high concentration levels of ~40 meq/l in the infiltrating water during both cycles of simulated rain events. These concentrations are much higher than those measured in the control experiment. As aforementioned, the high and relatively stable Ca and $SO_4$ concentration levels are indicative of the high content of phosphogypsum in the polluted sand, which effectively acts as a slow release source for both ions. In concur with the batch experiment results, and for the same reasons (some Ca-carbonate dissolution), the Ca concentrations were more stable than the $SO_4$ concentrations, which exhibited some decline with time (from ~40 to ~30 meq/l).

As detailed above, due to rapid NaCl dissolution and transport of the ions downward, only minor concentrations of Na and Cl were measured in the surface runoff water from the two simulated rain events (**Fig. 6A and 6B**). On the other hand, Ca and $SO_4$ concentrations in the runoff water from the first simulated rain event were high and similar to those measured in the infiltrating water. This indicates the presence of sufficient amount of gypsum at the soil surface to interact and dissolve in the surface water. However, in the second simulated rain event, both Ca and $SO_4$ concentrations at the surface runoff water were lower, probably due to substantial dissolution and removal of the gypsum from the very top levels of the soil profile during the first rain event.

Given the above detailed experimental results and observations, a similar process is expected under natural conditions, whereby the contaminates and solutes from the upper soil will be mobilized by the infiltrating rain water to the lower and deeper parts of the soil profile. This process was further studied in the field and by numerical modelling.

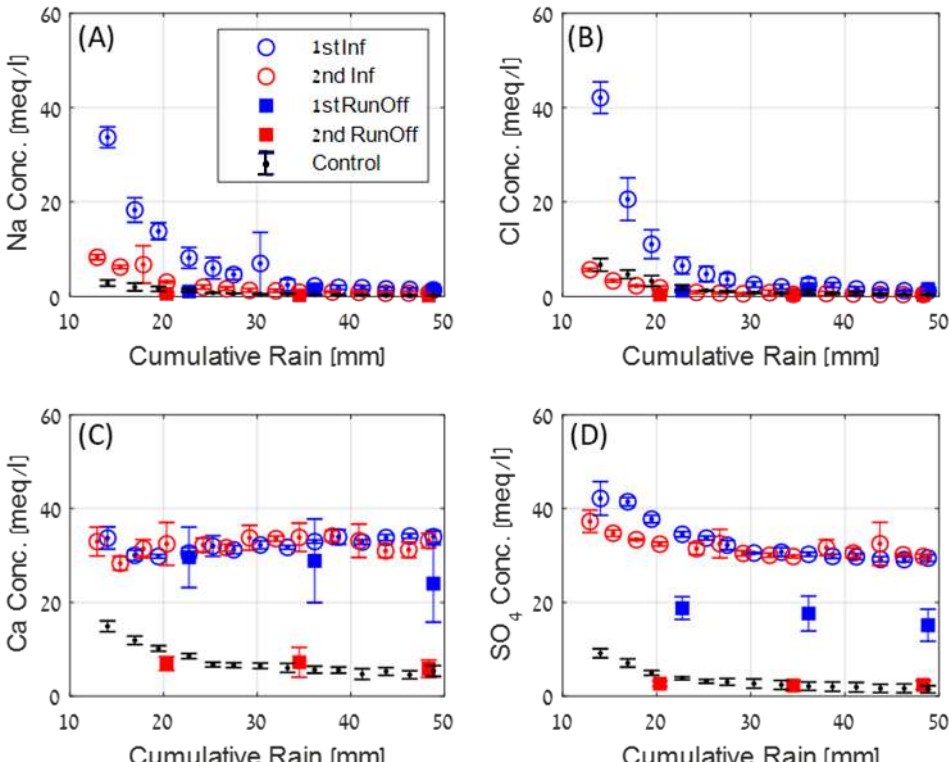

**Figure 6: Measured concentrations of Na (A), Cl (B), Ca (C), and SO₄ (D) in runoff water (squares) and infiltrating water (open circles) during two consecutive simulated rain events. Blue and red symbols are for first and second simulated rain events, respectively. Black symbols represent the measurements of the infiltrating water from the un-contaminated control soil. Error bars are from the four replicas.**

### 3.3 Field measurements - annual salt dynamic in sandy alluvial terraces

Temporal monitoring of the soil profiles in two sandy alluvial terraces was carried between the summers of 2020 and 2021. Major ions compositions along the soil profiles were characterized following extraction of the soluble salts with de-ionized water (1:1 dry soil : water ratio), as detailed above. The Cl and Ca measurements are hereafter presented and discussed, as these originate from two different salts, with markedly different solubilities and sources, namely halite and phosphogypsum, respectively. In addition, Cl is a conservative anion that does not interact with the soil, while the Ca cation may be absorbed to clay particles.

**Figure 7** present the results from the field study. On an annual scale, no change was found in Cl and Ca concentrations at the soil surface of the eastern plot between September 2020 and September 2021. Cl concentrations were on the order of 1.5 meq/l (**Fig. 7C-D**), and Ca concentrations were on the order of 37 meq/l (**Fig. 7E-F**). The difference in concentrations between Cl and Ca are in accordance with the laboratory experiments, reflecting the abundance of gypsum in the contaminated soils. In the western plot, no changes in Cl concentrations at the soil surface were observed between September 2020 and September 2021 (**Fig. 7G-H**). However, Ca concentrations declined by ~20% between September 2020 to September 2021 (**Fig. 7I-J**).

When examining the ions concentrations dynamics along the soil profiles in higher temporal resolution, between September 2020 and September 2021, it is evident that in both plots, during the winter months (October – March) Cl and Ca concentrations at the upper levels of the soil profiles were reduced. These concentrations drop are explained by dissolution of the salts and the leaching of the solutes downward with the infiltrating rainwater. Consequently, the concentrations of the ions were elevated at the subsurface, at depths of 30-40 cm, and 50-60 cm for the East and West Plots, respectively. These observations, point to

the dynamic nature of salts dissolution and solutes transport processes in the examined system. However, despite the high mobility of the solutes, under the arid conditions of the Ashalim Basin, the overall flushing and removal of the solutes and contaminants to significant depths is negligible. Apparently, the winter infiltration and leaching processes are not sufficient to mobilize the solutes deep enough to prevent their rise back by capillarity, during the long summer months, to the evaporation front at the soil surface.

As detailed above, the SEC model may be used to assess the effective depth from which contaminated pore water will not return by capillarity to the soil surface. The thickness of the SEC layer was calculated for the field site based on its hydrological properties (**Table 3**) and local climatic conditions (**Fig. 7 I-J**). The lower boundary of the SEC layer is depicted in **Fig. 7** by the red contours but its depth changes over the year due to temporal changes in the potential evaporation, $E_0$. In the east plot, the winter downward leaching of both Ca and Cl solutes did not pass the lower boundary of the SEC (**Fig. 7C, E**).

Consequently, during the dry summer months, from April onward, pore water and associated dissolved solutes were available for upward capillary flow towards the soil surface, where evaporation takes place. At the evaporation front, the solutes become concentrated in the solution and salts and minerals precipitate to form solid crusts at the soil surface (**Fig. 8**). **Figure 7D** illustrates a pronounced increase in chloride (Cl) concentration at the soil surface during the summer of 2021 compared to 2020, approximately doubling in magnitude. This enhancement is attributed to a thinner but more concentrated solute profile

near the soil surface in 2021 relative to 2020. As corroborated by **Figure 7C**, Cl accumulation at the end of the monitoring period in 2021 was confined to a ~10 cm layer, whereas in September 2020, it was distributed across a broader layer, ~20 cm. In the West plot however, some leaching of the solutes to depths of ~40 cm was recorded, yet no significant leaching of the solutes to depths greater than the SEC lower boundary were observed. The disparities between the East and West plots could be attributed to more efficient leaching processes that occurred in the West plot, due to slight differences in local meso-

topography and soil properties. However, throughout the entire period of measurements, in both plots, averaged measured $\theta_m$ along the soil profiles were on the order of 0.015, with maximal and rare values of 0.03, which were measured following the consecutive strong rain events of February 2021. These $\theta_m$ values correspond to $\theta_v$ of 0.025 and 0.05, respectively, and are consistently smaller than $\theta_{crit}$, which ranges between 0.15 in mid-winter and 0.2 during summer, as calculated by **Equation 5** for the measured hydraulic properties of the examined sand (**Table 3**) and computed $L_c$ (**Equation 1**). The fact that $\theta_v$ did not

exceed $\theta_{crit}$ supports other observations and understanding that there was no significant leaching of water and solutes from the SEC downward.

The data from the field sites demonstrate the dynamic nature of the solutes as they move vertically along the soil profile with wetting and drying events. In accordance with the SEC model, as long as $\theta_v < \theta_{crit}$ the solutes are not mobilized below the

lower boundary of the SEC layer and are expected to migrate upward during the dry summer months towards the evaporation front at the soil surface. As illustrated in **Figure 7**, Ca exhibits lower mobility compared to Cl within the soil profile, resulting in wider and deeper Ca distribution. This disparity can be attributed to the elevated Ca concentration in the contaminant effluent, the reduced solubility of gypsum relative to halite, and the propensity of Ca to be adsorbed onto soil clay particles. Presently (2024), more than six years since the contamination event, salt crusts, mainly of phosphogypsum, are still widely seen in the contaminated basin (**Fig. 8**). For additional evaluation and understanding of water flow processes and solute transport dynamics, a numerical model was used to examine the system, as detailed below.

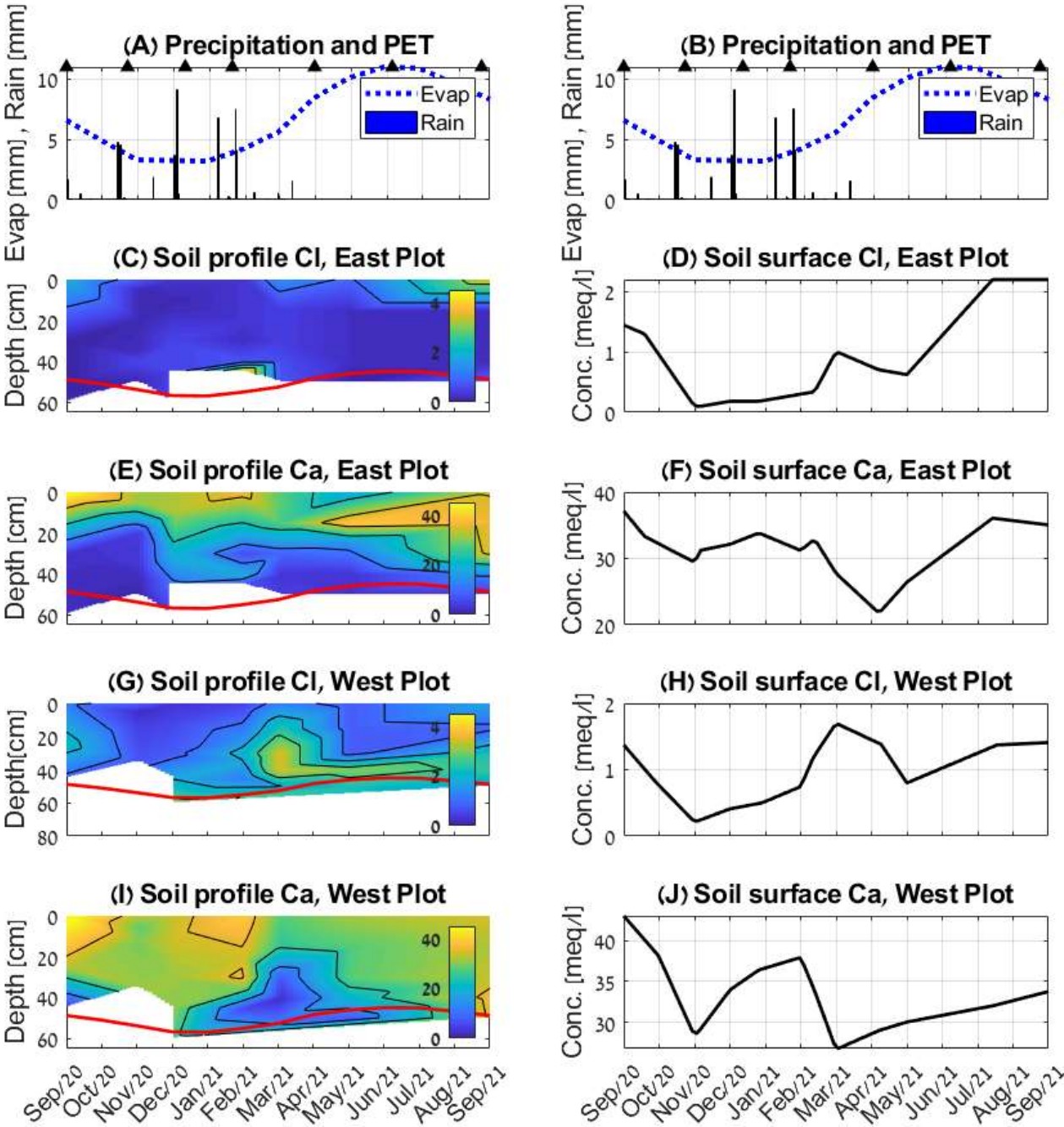

**Figure 7:** Measured precipitation and potential evaporation (PET) are presented in (A-B). Triangles indicate time of soil sampling. Temporal changes in Ca and Cl depth profiles in the sandy terraces, as measured at the East (C-F) and West Plots (G-J). The left panels present Cl and Ca concentrations (meq/l) over the entire soil profiles, down to the maximal depth that was sampled. The red contours mark the lower boundary of the SEC layer. The right panels present ions concentrations at the soil surface only.

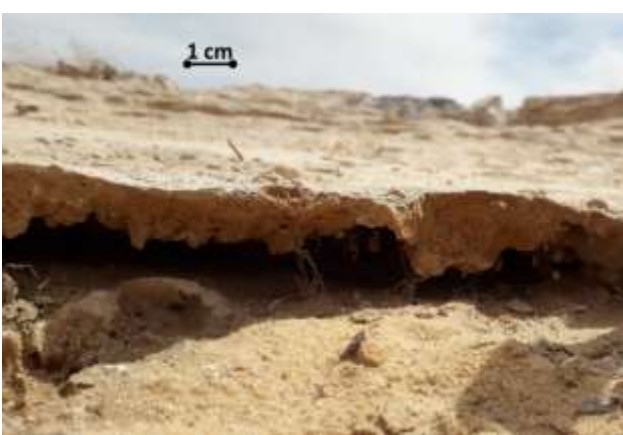

**Figure 8: Thin and brittle crust of gypsum that formed over the contaminated sandy soils.**

### 3.4 Numerical simulation model

380 The numerical model presents solute concentration levels (**Fig. 9C, E, G**), and water flow directions and velocities (**Fig. 9D, F, H**) along a 100 cm soil profile over a time period of 12 months. The solute simulated in the model is considered conservative, implying that it does not undergo sorption onto soil particles, similar to the behaviour of Cl. Precipitation and evaporation conditions measured at the field site between September 2020 and September 2021 (**Fig. 9A-B**) were used to define the upper boundary condition for the reference simulation. Additional two simulations were done for increased precipitation by two and

385 five folds.

Model results for the actual rain and evaporation conditions demonstrate that the removal of solutes to depths greater than 30 cm is negligible. It is seen in **Fig. 9C** that for the examined setup and typical climatic conditions, the solutes are being leached downward during and following rain events, whereas in the following dry periods they are being transported upward by rising capillary water towards the evaporation front at the soil surface. Therefore, and in agreement with field measurements, mainly

390 for Cl, over a full winter-summer cycle, the high concentration of the solutes at the upper levels of the soil profile is maintained. **Figure 9D** presents water flow directions and velocities along the soil profile, where negative values (bluish colours) indicate downward infiltration, and positive values (yellowish colours) indicate upward capillary flow. It is seen that for the prevailing climatic conditions, which included only three rain events, a notable downward transport of the solutes to depths of about 20 cm took place. Each rain event was followed by a reversed water flow direction that occurred immediately after the rain event

395 ended (**Fig. 9D**). This is in accordance with field measurements and the SEC model, with its lower boundary marked by the red contours in **Fig. 9**. As aforementioned, solutes that do not leach to depths greater than this SEC lower boundary are available for upward capillary flow towards the evaporation front, at the soil surface.

In the simulated conditions of doubled rain depth, during the two rain events (December 2020, and February 2021) the maximal depth of infiltrating water approaches the lower boundary of the SEC layer (**Fig. 9F**), and consequently, a minor reduction in

400 solutes concentration at the upper levels of the soil profile is simulated. For the extreme simulated conditions of five times

greater rain depth than the natural conditions, the depth of the infiltrating water is much deeper than the SEC depth (**Fig. 9H**), and indeed the upper levels of the soil profile are completely depleted of the solutes (**Fig. 9G**). The differences in water flow and solute leaching dynamics between the three simulated precipitation conditions are in line with simulated water content of the SEC and estimated $\theta_{crit}$. As detailed in section 1.2, $\theta_{crit}$ denotes the minimal level of water content at the SEC that is needed

to support water infiltration and solute transport to depths greater than the lower boundary of the SEC. **Figure 10** presents averaged water content of the SEC, based on simulated water content levels at depths of 0, 10, 20, 30, 40, and 50 cm. For the actual rain conditions, as monitored in the field, the simulated (and measured) maximal SEC water content, following the strong rain event of December 2020, was at the order of 0.05, much lower than $\theta_{crit}$. Consequently, water flow below the SEC was not observed in the simulation. For the doubled rain scenario, maximal SEC water contents were elevated to levels of

about 0.07, which is still below $\theta_{crit}$. It is only in the extreme simulated climatic conditions of increased precipitation by five folds, that the SEC averaged water content almost approaches 0.15, and a substantial leaching of solutes from the SEC is modeled. **Figure 10** also presents averaged water content levels at the SEC, as measured in the field for the East and West plots. In agreement with the numerical simulation, it is seen that water content levels in the field were much lower than the needed $\theta_{crit}$ of 0.15, to support downward infiltration from the SEC.


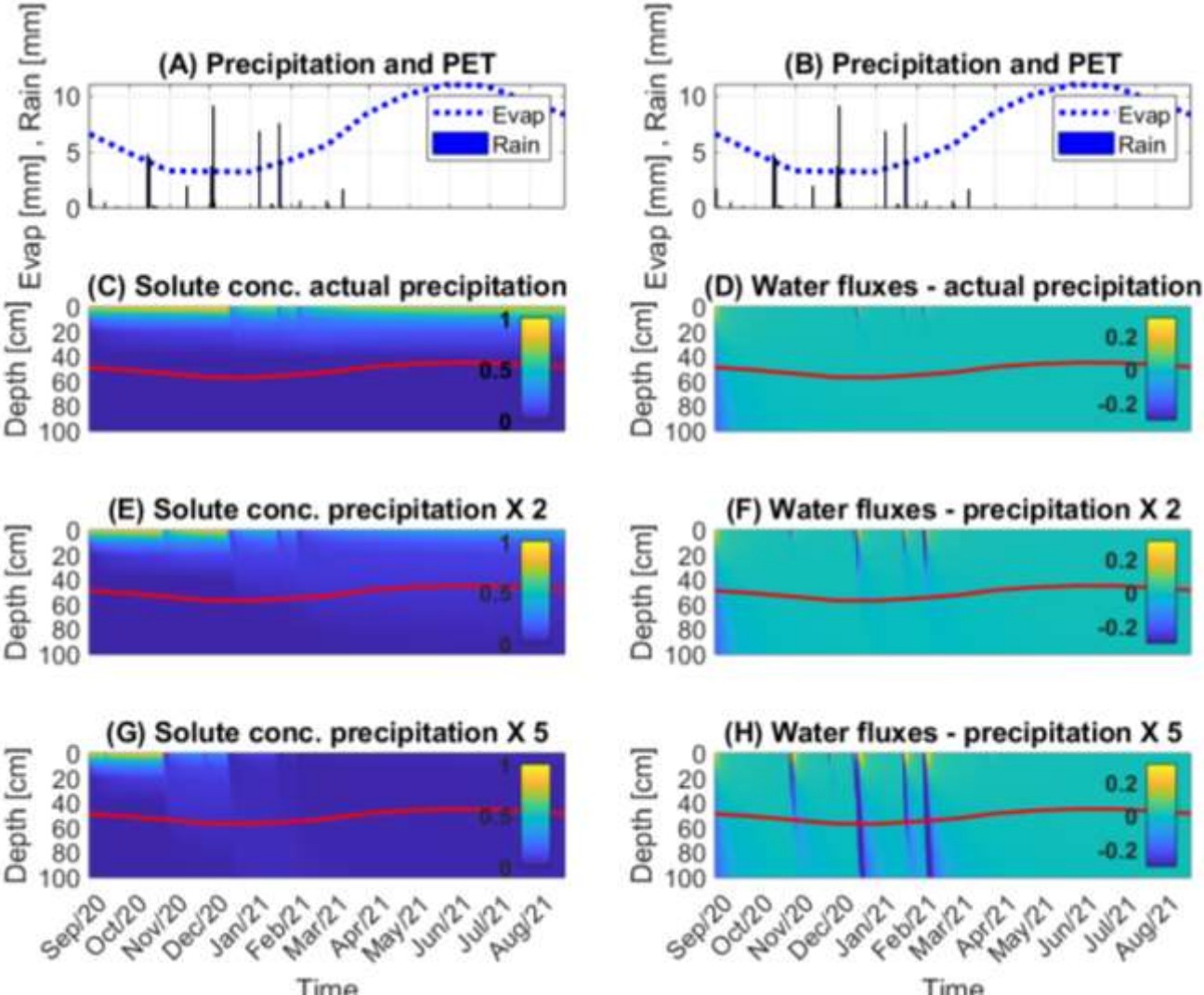

**Figure 9: Numerical simulation results for solute concentration (left panels) and water fluxes (right panels) for three precipitation scenarios. (A) and (B) present the measured precipitation and potential evaporation (PET) conditions used for the simulation. (C, D) present computed solute concentration and water flow velocities, respectively, for actual climatic conditions. (E, F), and (G, H), present the same properties, for two and five folds elevated precipitation, respectively. Positive values in (B, D, F, H) indicate upward water flow. Red line signifies the calculated characteristic evaporation length of the soil, $L_C$, which is the lower boundary of the SEC.**


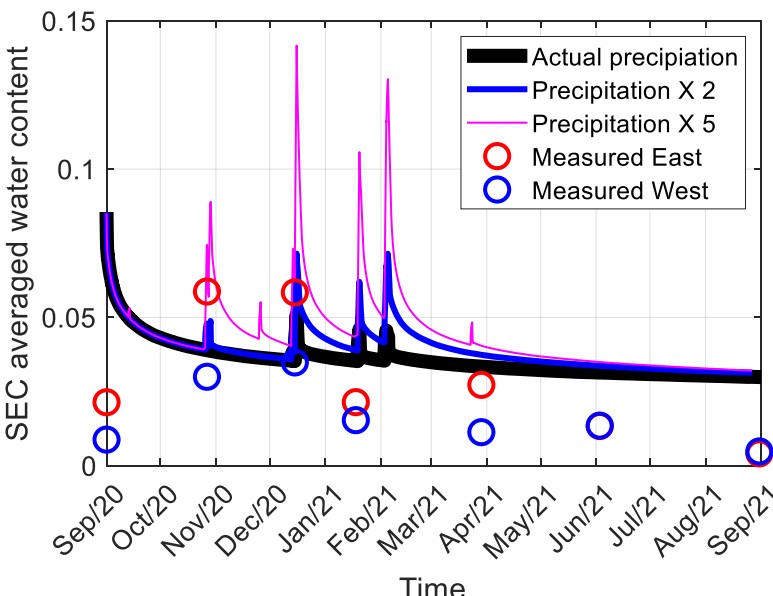

**Figure 10: Averaged SEC water content over time for numerical simulations (solid curves) and field measurements at the East and West plots (circles).**

**3.5 Applying the SEC model to predict the potential of solutes and contaminants leaching from different soil textures**

Following the experimental results, field measurements and the numerical model presented above, it is hereby proposed to use the SEC model as a tool to estimate solutes and contaminants transport and accumulation along the soil profile, for non-reactive

elements. This will be done by estimating the net amount of water entering the SEC and its impact on the average water content of the SEC, which affects water flow and solutes transport processes along the soil profile. For a given cumulative precipitation and evaporation conditions, over a certain time period, the water content of the SEC ($\theta_{SEC}$) can be calculated by dividing the net length of water entering the SEC, by the SEC depth (i.e. $L_C$ (mm)):

$$\theta_{SEC} = \frac{(P-E)}{L_C} \tag{8}$$

where $P$ (mm) and $E$ (mm) are precipitation and evaporation, respectively, over the examined time period. As aforementioned, in order to initiate water flow through the SEC lower boundary, $\theta_{SEC}$ need to be equal to or higher than $\theta_{crit}$. Consequently, by substituting $\theta_{SEC}$ by $\theta_{crit}$ in **Equation 8** and rearranging the equation, it is possible to depict the required $P$, for a given $E$, to allow water flow and leach of solutes and contaminates out of the SEC:

$$P = \theta_{crit} \cdot L_C + E \tag{9}$$

where $\theta_{res}$ and $\theta_{sat}$ are residual and saturated water contents, respectively, and $m$ is derived from the van Genuchten $n$ parameter (**Equation 3**). **Figure 11** presents computed $L_C$ , based on **Equations 1-4**, for various soil textures and five daily evaporative scenarios. **Table 4** presents the physical and hydraulic properties of the examined soils, used for the calculations, as taken from the HYDRUS-1D library (Schaap,2001).

In agreement with previous calculations (Or and Lehmann, 2019), the loamy soils have the deepest $L_C$ (**Figure 11A**). For
coarser texture soils, $L_C$ is shorter due to limited pore size distribution and minor capillary gradient. For finer texture soils, high viscous resistance and low hydraulic conductivities also lead to short $L_C$ (Or and Lehmann, 2019). Moreover, it is seen in **Figure 11A** that for the relatively high evaporative levels (daily evaporation of 7-13 mm), $L_C$ is quite short and with small differences between the different soil types. Since high evaporation rates usually characterize the dry season, especially in arid areas, (see **Fig's 7 and 9**), there is no practical influence of these short $L_C$ values on solutes and contaminants leaching.
However, for low evaporative levels (daily evaporation of 1-4 mm), that are commonly associated with rainy periods, a notable difference in $L_C$ length is observed between the different soil types. **Figure 11B** demonstrates that relatively low $P$ levels are sufficient to drain the SEC of fine texture soils, due to the very short $L_c$ of such clayey soils. Highest $P$ levels are needed to drain the SEC of the middle texture and leach down its solutes and contaminants. This is a result of the long $L_c$ of soils such as loamy soils, mainly at the low evaporation rates conditions. Shorter $P$ depth are also needed for the drainage of SEC of
sandy soils, due to the low $\theta_{crit}$ values and relatively short $L_c$ of these soils.

**Table 4: Soil textural and physical properties from HYDRUS-1D library, used for calculations of $K_{eff}$ and $\theta_{crit}$, (in table), and $L_C$ and $P$, (in Figure 11).**

| Soil texture | Residual water content $\theta_{res}$ [-] | Saturated water content $\theta_{sat}$ [-] | van Genuchten parameters | | | Saturated hydraulic conductivity $K_S$ [mm/day] | SEC effective hydraulic conductivity $K_{eff}$ [mm/d] | water content needed to drain SEC $\theta_{crit}$ [-] |
|---|---|---|---|---|---|---|---|---|
| | | | $\alpha$ [1/mm] | $n$ [-] | $m$ [-] | | | |
| Sand | 0.045 | 0.43 | 0.0145 | 2.68 | 0.626866 | 7128 | 79.40 | 0.21 |
| Loamy Sand | 0.057 | 0.41 | 0.0124 | 2.28 | 0.561404 | 3502 | 32.54 | 0.23 |
| Sandy Loam | 0.065 | 0.41 | 0.0075 | 1.89 | 0.470899 | 1061 | 7.153 | 0.31 |
| Loam | 0.078 | 0.43 | 0.0036 | 1.56 | 0.358974 | 249.6 | 0.957 | 0.42 |
| Silt | 0.034 | 0.46 | 0.0016 | 1.37 | 0.270073 | 60 | 0.117 | 0.46 |
| Silt Loam | 0.067 | 0.45 | 0.002 | 1.41 | 0.29078 | 108 | 0.254 | 0.45 |
| Sandy Clay Loam | 0.1 | 0.39 | 0.0059 | 1.48 | 0.324324 | 314.4 | 0.958 | 0.38 |
| Clay Loam | 0.095 | 0.41 | 0.0019 | 1.31 | 0.236641 | 62.4 | 0.087 | 0.41 |
| Silt Clay Loam | 0.089 | 0.43 | 0.001 | 1.23 | 0.186992 | 16.8 | 0.0125 | 0.43 |
| Sandy Clay Loam | 0.1 | 0.38 | 0.0027 | 1.23 | 0.186992 | 28.8 | 0.0215 | 0.38 |
| Silty Clay | 0.07 | 0.36 | 0.0005 | 1.09 | 0.082569 | 4.8 | 0.00030 | 0.36 |

| Clay | 0.068 | 0.38 | 0.0008 | 1.09 | 0.082569 | 48 | 0.0030 | 0.38 |
|------|-------|------|--------|------|----------|-----|--------|------|

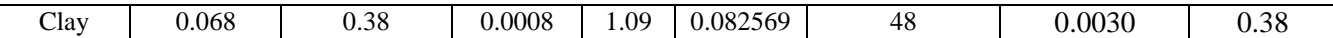

**Figure 11: The impact of different evaporative conditions and soil textures on $L_C$ (A), and minimal amount of daily precipitation, $P$, needed to initate drainage from the SEC (B).**

## 4 Summary and conclusions

Experimental results, field measurements, and numerical model simulations all indicate that under arid conditions and environments, the mobilization and removal of solutes and pollutants to substantial depths in sandy soils is a minor process even when considering solutes that do not tend to be absorbed to soil particles. Given the relatively high hydraulic conductivities of the sandy soils it is apparent that similar conclusions can be drawn for heavier and less permeable soils under arid conditions. The sporadic rain events that occur in arid environments enable only minor downward mobilization of the solutes, whereas the dry periods between such rain events are sufficient to force capillary upward transport of the solutes and dissolved pollutants. **Figure 12** is a conceptual model that illustrates the dynamics of water flow and solute transport processes, as demonstrated in this work. The soil evaporation capacitor (SEC) model (Or and Lehmann, 2019) which describes the depth of soil that contributes water for the first stage of evaporation, was applied to estimate the depth to which the solutes and contaminates circulate vertically, further supporting the conceptual model.

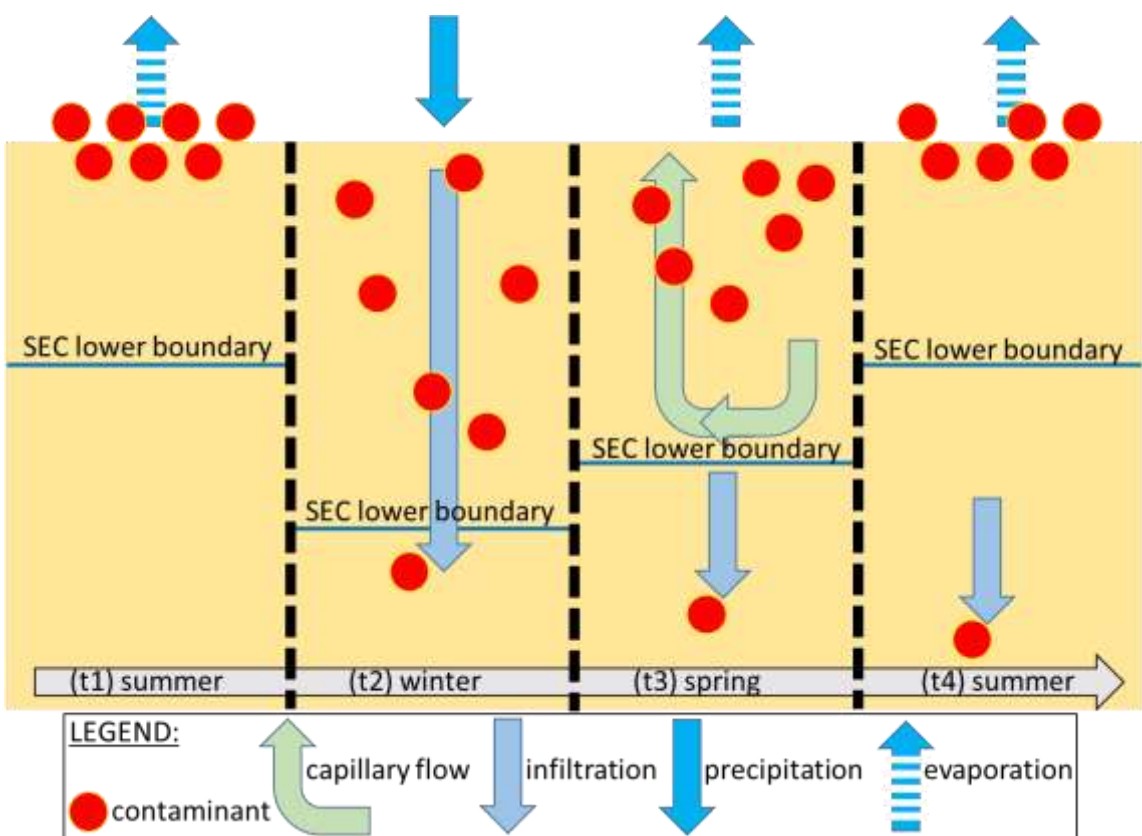

**Figure 12: Conceptual model describing the fate of solutes along a soil profile under arid environments following the SEC concept. During rain events the solutes are being leached downward, but as long as they remain within the SEC zone, they can move upward with capillary water, towards the evaporation front at the soil surface. Solutes that are mobilized to depths greater than the SEC lower boundary can no longer flow upward towards the soil surface.**

Since catastrophic pollution events in arid environments are in many cases of extreme magnitude, and related to extreme flash flood events, it is likely that vast areas beyond the MFC will be exposed to the contamination. These areas, however, do not experience regular and natural flash floods, and thus their natural attenuation processes driven by water are expected to be very limited. Therefore, it is estimated that areas like the sandy terraces of the Ashalim stream will remain polluted long after the spill event, imposing significant chemical stress to the habitat and rendering it to slow remediation. During this period, it may also act as a slow release source of contamination to the environment.

Based on the various observations and analyses presented in this study, it is proposed to apply the SEC model as a practical tool for estimating the potential mobility of solutes and contaminates from the upper part of the soil profile, for different soil types and climatic conditions. The soil textural properties and evaporative conditions determine the SEC depth, and the minimal water content of the SEC soil that is needed to initiate effective drainage and solutes removal from the SEC to deeper horizons of the soil profile. Therefore, simple and basic information about the soil properties and climatic conditions of a

contaminated area may be sufficient to provide an estimate as to the removal efficiency of the contaminated layer by leaching processes, a critical information when considering bioavailability of contaminants.

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
