# Peer review of "Soil contamination in arid environments and assessment of remediation applying surface evaporation capacitor model; a case study from the Judean Desert, Israel"

_EGUsphere, 2024_

## Author Comment (AC2)

The article sheds light on contaminant transport in arid region using a combination of lab and field experiments and numerical simulations. The incident releasing contaminants in the Judean desert was a flash flood after the breaking of a dike in the year 2017. After this event, salts and contaminants are redistributed in the soil profile during rainfall infiltration and evaporation. The authors apply the surface evaporation capacitor concept to test if contaminants could percolate to deeper soil layers and are removed from the active evaporation layer.

The topic, the case study, and the applied methods are interesting, but the analyses must be extended and presented in more detail as explained below.

We thank the reviewer for the positive and constructive comments. We addressed each comment raised by the reviewer, please see replies below (in red).

1. surface evaporation capacitor: the percolation from the capacitor to deeper soil layers depends on the water content of the capacitor. The water content in the capacitor must be higher than the critical water content (as calculated in Assouline and Or, 2014, WRR WR015475, or Lehmann et al, 2019, GRL GL083932). The authors should expand the SEC-analysis by estimating the water content after the winter rainfall events to check if percolation to deeper soil layers can occur (for the calculated thickness of the capacitor, what is the water content after a certain rainfall event?).

This is a good point. Indeed, percolation below the capacitor will initiate once water content at the SEC goes beyond a critical water content value - $\theta_{crit}$. As detailed in the works mentioned by the reviewer and others, $\theta_{crit}$ is soil water content retained at the soil for suction pressure $L_c$ , that describes the soil characteristic evaporation length ($h_{crit}=L_c$). As stated by Lehmann et al (at various papers) water at depths greater than $L_c$ will not be exposed to evaporation during first stage of evaporation and it may drain downward.

Once again - if water content at the SEC will go over $\theta_{crit}$ – water leakage from the SEC will start.

For calculated $L_c$ in our work (as appears in Figures 8 & 9, by the red contour), and based on the VG model and the hydraulic parameters as appear in Table 3, average $\theta_{crit}$ is equal to 0.1776 with maximal values during mid-summer (=0.197) and minimal values during mid-winter (=0.1561). This means that for water contents higher than ~0.15, during winter, it is expected that water will drain from the SEC downward.

Mass soil water content measurements were done during winter after each rain event (1-2 days following the rain event). Measurements were done by sampling the moist soil from various depths, weighing of the wet soil, oven drying of the soil samples at 105°C (for 24hr), and weighing of the dry soil. Averaged measured mass water content was at the order of 0.02, with maximal values of 0.06 (rarely measured). For soil bulk density of ~1.67 (g/cm$^3$), that is computed for the sandy soil at the site, upon its measured porosity, the equivalent volumetric water content is at the order of 0.033 which is much smaller than $\theta_{crit}$ , hence it is likely to assume that leakage from the SEC is negligible at the test site, under the examined conditions.

This was discussed in the revised paper, where at the introduction we explained the relation between water content and leakage from the SEC (P5, L121-122). At the M&M section we elaborated about the procedure of water content measurements (P8, L187-194), and at the results and discussion section, we discussed it further, considering measured water content values (P15, L344-350).

2. Hydrus-1D simulations: After the simulation of the time period Sept_20 to August_21 presented in figure 9, the solute is concentrated in soil layers close to the surface and is not redistributed to larger depths shown in the experimental findings (figure 7). This discrepancy can be partially related to the flash flood that cannot easily be simulated. Another effect that should be taken into account in the simulation is the repeated redistribution between the incident 2017 until 2021. How is the solute plume travelling with depth for this 4-5 years period?

For simplicity, and in order to clearly visualize and characterize the simulated solute transport processes and to correlate it to the SEC concept, the initial conditions of the model were set with contamination to be located only at the top10 cm of the soil profile.

The effect of flash flood is not being taken into account as it is assumed (and observed at the site) that the sandy terraces are above the mainstream flow channel and are not being flooded during normal and natural flash flood. The only (natural) sources of water that the terraces are being exposed to are the sporadic and minor rain events that may occur during winter. These rain events are included in the model. The flood of the catastrophic contamination event was much higher than the terraces, therefore they were highly polluted.

The initial conditions of the model are quite similar to the measured conditions of Cl concentration at Sep. 2020, as at this time Cl concentration was concentrated at the top 10 cm of the East plot and ~20 cm of the west plot. For Ca however, the solutes concentration distribution was wider to depths of ~30cm. This disparity is associated to the much higher quantities of calcium at contaminated effluent, and in particular, the fact that Ca may go through processes of absorption to the soil particles and it is less mobile than Cl. This is already mentioned in the text, yet we elaborated on that at the revised paper at the field measurement section (P15-16, L354-357) and the numerical model section (P17, L370-371).

3. Soil water retention: The soil water retention curve was measured with the hanging column method resulting in shape parameters of alpha equal to 0.011/cm and n = 2.8 (Table 3). With such a small alpha the drainage occurs between 50 and 200 cm. Was this pressure range covered with the hanging water column method? The authors should show both the measured values and the fitted curve.

The hanging column test went down to suction of 160 cm. Soil drainage occurred in between the suctions of 20 to 60 cm. At suction of ~150cm water content of the sample was already equal to the residual water content and no more water went out of the sample. The WRC data is added to the data presented at the Repository.

4. Saturated hydraulic conductivity: In contrast to the soil water retention curve, the saturated conductivity was not measured but estimated with Rosetta implemented in Hydrus-1D. The predicted value for a rather dense packing of ~1.7 g/cm3, is about 25

cm per day; the other predicted parameters (probably about n=1.41 and alpha = 0.0268/cm) are quite different compared to the lab experiments (n = 2.8 and alpha = 0.011/cm). The combination of parameters obtained with different approaches in the SEC-model may lead to inconsistent values of the thickness of the capacitor layer. In addition, the predicted hydraulic conductivity is rather small compared to the irrigation rate applied in the rain simulator experiments (48 mm per hour or 115.2 cm/day). I would expect that a saturated hydraulic conductivity much smaller than the irrigation rate would result in more runoff than found in the experiments presented in Figure 5.

This is a very good point. Thanks to this comment we did a Darcy test to determine saturated hydraulic conductivity of the soil. Indeed, as speculated by the reviewer, Ks is much higher and it is at the order of 250 cm/d. We repeated all simulations with this value. Moreover, we did more repetitions for the hanging column test and we refined the V.G. parameters (not a big difference from previous values). Now all physical properties of the soil were determined based on physical measurements and the numerical simulations were changed accordingly. Fortunately, these changes did not change the processes, mechanisms and trends, which were discussed and examined in the paper. In the M&M section we mentioned that Ks was measured by a Darcy test (P6, L140-141).

5. Figure 7, there is an increase of chloride in the East Plot. What are the hypotheses for that increase? Could you add in figures D, F, H and J the concentrations in deeper layers as well (line for 30-40 or 50-60 cm according to line 315). Please provide more information on the rainfall rates and amounts and on the profile measurements (show in the figure when the samples were collected). Do you expect identical hydraulic properties in East and West plot? Could this be tested?

Unfortunately, it is impossible to present deeper information. We presented all available data. In part of the samplings, we managed to reach deeper levels than others. Complete information about rainfall and evaporation is presented in panels A and B. As mentioned by the figure caption, the triangles at the upper X axis of panels A and B present times of soil sampling. Nothing is identical in nature, but it is believed / assumed that the sands in both terraces are alike. Texture measurements, which were done for samples from both sites support this assumption.

As for the first comment regarding increase in Cl concentration: this could be attributed to upward flow of Cl from levels deeper than ~60 cm, which were not sampled in Sep. 2020. However, this is not very likely as these depths are greater than the lower boundary of the ESC. We believe that the net change in solutes mass was negligible and that the reason for changes in concentrations is due to accumulation of the solutes in narrower layer in 2021. It is seen that in Sep. 2020 the "salt bulb" went down to depth of ~20 cm, whereas at Sep. 2021 it was concentrated in a narrower layer with a thickness of ~10 cm. In practice, it means that in 2021 the concentration is ~2 times higher than measured concentrations in 2020. This is mentioned at the revised manuscript (P15, L337-339).

[revised manuscript text omitted]

---

## Author Response (AR1)

Dear editor,

We wish to thank the two reviewers for their positive review and constructive comments. Below are the replies for each one of the reviewer's comments.

Thanks to the reviewers we elaborated more on the SEC model and better explained the needed physical conditions to initiate drainage from the SEC. In the revised manuscript we explain the term 'critical water content', that is the minimal needed water content of the SEC, which allows water infiltration downward. In addition, another subsection was added to the results and discussion section, where we propose to use the SEC model as a tool to predict the potential of solutes and contaminants leaching from different soil textures.

We hope you will find the revised paper suitable for publication in *SOIL*.

**REVIEWER #1:**

The article sheds light on contaminant transport in arid region using a combination of lab and field experiments and numerical simulations. The incident releasing contaminants in the Judean desert was a flash flood after the breaking of a dike in the year 2017. After this event, salts and contaminants are redistributed in the soil profile during rainfall infiltration and evaporation. The authors apply the surface evaporation capacitor concept to test if contaminants could percolate to deeper soil layers and are removed from the active evaporation layer.

The topic, the case study, and the applied methods are interesting, but the analyses must be extended and presented in more detail as explained below.

We thank the reviewer for the positive and constructive comments. We addressed each comment raised by the reviewer, please see replies below (in red).

1. surface evaporation capacitor: the percolation from the capacitor to deeper soil layers depends on the water content of the capacitor. The water content in the capacitor must be higher than the critical water content (as calculated in Assouline and Or, 2014, WRR WR015475, or Lehmann et al, 2019, GRL GL083932). The authors should expand the SEC-analysis by estimating the water content after the winter rainfall events to check if percolation to deeper soil layers can occur (for the calculated thickness of the capacitor, what is the water content after a certain rainfall event?).

This is a good point. Indeed, percolation below the capacitor will initiate once water content at the SEC goes beyond a critical water content value - $\theta_{crit}$. As detailed in the works mentioned by the reviewer and others, $\theta_{crit}$ is the soil water content retained in the soil under suction pressure $L_c$ , that describes the soil characteristic evaporation length ($h_{crit}=L_c$). As stated by Lehmann et al (in an number of papers) water at depths greater than $L_c$ will not be exposed to evaporation during first stage of evaporation and may drain downward.

Once again - when water content at the SEC exceeds $\theta_{crit}$ – water leakage from the SEC is expected to occur.

For the calculated $L_c$ in our work (as appears in Figures 8 & 9, by the red contour), and based on the VG model and the hydraulic parameters in Table 3, our average $\theta_{crit}$ is 0.1776 with a maximal value during mid-summer (=0.197) and minimal value during mid-winter (=0.1561). This means that during winter, for water contents higher than ~0.15, water is expected to drain from the SEC downwards.

Soil water contents were measured during winter after each rain event (1-2 days following the rain event). Moist soil samples were collected from various depths, weighied, oven dried at 105°C (for 24hr), and re-weighed as dry soil. Average measured mass water content was ~0.02, with maximal values of 0.06 (rarely measured). The computed bulk density of the sandy soil at the site is ~1.67 (g/cm³), and given its independently measured porosity, the equivalent volumetric water content is around 0.033, which is much smaller than $\theta_{crit}$. Hence it is reasonable to assume that under the conditions at the study site, leakage from the SEC is negligible.

Following the reviewer's comment, the above explanations and clarifications were added to the revised paper: In the Introduction we now explain the term $\theta_{crit}$ , and the relation between water content and leakage from the SEC (p. 5, L124-127). In the Material and Method section we elaborate about the procedure of water content measurements (P8, L191-199), and in the Results and Discussion section, we discuss it further while considering the measured water content values (P15, L355-361). $\theta_{crit}$ is also being referred in the results of the numerical model section (Section 3.4) and another sub-section (3.5) entitled '*Applying the SEC model to predict the potential of solutes and contaminants leaching from different soil textures*' was added at the end of the results and discussion section. In this section we discuss the needed condition to imitate SEC drainage for different soil texture properties and evaporative conditions.

2. Hydrus-1D simulations: After the simulation of the time period Sept_20 to August_21 presented in figure 9, the solute is concentrated in soil layers close to the surface and is not redistributed to larger depths shown in the experimental findings (figure 7). This discrepancy can be partially related to the flash flood that cannot easily be simulated. Another effect that should be taken into account in the simulation is the repeated redistribution between the incident 2017 until 2021. How is the solute plume travelling with depth for this 4-5 years period?

For simplicity, and in order to clearly visualize and characterize the simulated solute transport processes and to correlate it to the SEC concept, the initial conditions of the model were set with contamination to be located only at the top10 cm of the soil profile.

The effect of flash flood is not being taken into account as it is assumed (and observed on site) that the sandy terraces are above the mainstream flow channel and are not being flooded during normal characteristic natural flash floods. The main (natural) sources of water that the terraces are being exposed to are the sporadic and minor rain events that may occur during winter. These rain events are included in the model. The flood level of the catastrophic contamination event was much higher, covering and polluting the studied terraces.

The initial conditions of the model are quite similar to the measured conditions of Cl concentration on Sep. 2020; at that time Cl was concentrated at the top 10 cm of the East

plot and the top ~20 cm of the west plot. For Ca however, the solutes concentration distribution was wider, reaching depths of ~30cm. This disparity is attributed to the much higher concentration of calcium in contaminated effluent and slurry, and in particular, the fact that Ca may go through processes of precipitation or absorption to the soil particles and is therefore less mobile than Cl. This was already mentioned in the original text, but we emphasized it in the revised paper at the Field Measurement section (P17, L365-368) and the Numerical Model section (P17, L381-382) where we further elaborate on it.

3. Soil water retention: The soil water retention curve was measured with the hanging column method resulting in shape parameters of alpha equal to 0.011/cm and n = 2.8 (Table 3). With such a small alpha the drainage occurs between 50 and 200 cm. Was this pressure range covered with the hanging water column method? The authors should show both the measured values and the fitted curve.

The hanging column test went down to suction of 160 cm. Soil drainage occurred in between the suctions of 20 to 60 cm. At suction of ~150cm water content of the sample was already equal to the residual water content and no more water came out of the sample. To clarify this point, the WRC data was added to the data presented in the Repository section.

4. Saturated hydraulic conductivity: In contrast to the soil water retention curve, the saturated conductivity was not measured but estimated with Rosetta implemented in Hydrus-1D. The predicted value for a rather dense packing of ~1.7 g/cm3, is about 25 cm per day; the other predicted parameters (probably about n=1.41 and alpha = 0.0268/cm) are quite different compared to the lab experiments (n = 2.8 and alpha = 0.011/cm). The combination of parameters obtained with different approaches in the SEC-model may lead to inconsistent values of the thickness of the capacitor layer. In addition, the predicted hydraulic conductivity is rather small compared to the irrigation rate applied in the rain simulator experiments (48 mm per hour or 115.2 cm/day). I would expect that a saturated hydraulic conductivity much smaller than the irrigation rate would result in more runoff than found in the experiments presented in Figure 5.

This is a very good point. Thanks to this comment we did a Darcy test to determine saturated hydraulic conductivity of the soil. Indeed, as speculated by the reviewer, Ks is much higher and it is at the order of 250 cm/d. We repeated all simulations with this value. Moreover, we did more repetitions for the hanging column test and we refined the V.G. parameters (not a big difference from previous values). Now all physical properties of the soil were determined based on physical measurements and the numerical simulations were changed accordingly. However, these changes did not change the dominant processes, mechanisms and trends, which were discussed and examined in the paper. In the M&M section we mentioned that Ks was measured by a Darcy test (P6, L146-147). The reported results were corrected accordingly.

5. Figure 7, there is an increase of chloride in the East Plot. What are the hypotheses for that increase? Could you add in figures D, F, H and J the concentrations in deeper layers as well (line for 30-40 or 50-60 cm according to line 315). Please provide more information on the rainfall rates and amounts and on the profile measurements (show in the figure when the samples were collected). Do you expect identical hydraulic properties in East and West plot? Could this be tested?

Unfortunately, it is impossible to present deeper information. We presented all available data. In part of the samplings, we managed to reach deeper levels than others. Complete information about rainfall and evaporation is presented in panels A and B. As noted in the figure caption, the triangles at the upper X axis of panels A and B present times of soil sampling. The natural sites are not homogeneous and the two sites are not identical but we attempted to choose similar sites, in close proximity. Our initial impression on site, indicated they are both representative of the area, and not so different from one another. The main sedimentary sources are driven from the Hazeva sandy formation, mixed with local carbonate rock fragments and loess. Yet, unavoidably, there are slight variations between sites and even in the same site at short distance apart. Soil texture measurements, which were done for samples from both sites support this. Yet, these differences do not have significant impact on our conclusions.

As for the first comment regarding increase in Cl concentration: this could be attributed to upward flow of Cl from levels deeper than ~60 cm, which were not sampled in Sep. 2020. However, this is not very likely as these depths are greater than the lower boundary of the ESC. We believe that the net change in solutes mass was negligible and that the reason for changes in concentrations is due to accumulation of the solutes in narrower layer in 2021. It is seen that in Sep. 2020 the "salt bulb" went down to depth of ~20 cm, whereas at Sep. 2021 it was concentrated in a narrower layer with a thickness of ~10 cm. In practice, it means that in 2021 the concentration is ~2 times higher than measured concentrations in 2020. This is mentioned at the revised manuscript (P16, L347-351).

REVIEWER #2:

This paper presents an interesting study on the downward movement of solutes in arid environments. Experimental data are compared with model simulations and with the surface evaporation capacitor SEC concept. Especially the testing of the SEC concept based on numerical model simulations could make this a very interesting paper. I think that the authors should improve on this test to make it impactful. In the current state of the paper, the test is too vague to show the potentials of the SEC. I made a suggestion on how this test could be improved. Another point of concern I have is that a very simple indicator or parameter for solute transport, namely the difference between precipitation and evaporation is not considered to explain the observed behavior of the solute accumulation near the soil surface and the solute leaching. Only when this difference is zero or negative (e.g. when there is a long-term net upward flow from for instance a shallow groundwater table), there is no leaching or downward movement. I think the authors should also give information about this parameter during the monitoring period. Especially since downward movement of solutes is actually used to estimate net groundwater recharge (or the difference between precipitation and evaporation) in dry environments, this is important to address.

The important general summary presented above by the Reviewer is also detailed in the comments that follow, mostly the last three comments. We address these together at the end of our replies. These comments motivated us to add a new section to the discussion thereby greatly contributing to the paper, for which we are very thankful.

Detailed comments:

Line 25: Are semi-arid regions also deserts?

Indeed, semi-arid region is not a desert therefore this phrase was deleted.

Ln 38: are increased flood probabilities due to climate change also predicted for deserts?

Yes. Relevant references are cited in the original manuscript, and another one was added (Sen et al., 2013).

Ln 43-53: Can you give some discussion why it is important to give attention to the contamination of the terraces since I suspect that the contaminants are also present in the MFC. What do we know about the contamination of the MFC and why can't we extrapolate that knowledge to the terraces?

Done. the following text was added: "While a contaminated MFC is expected to experience natural processes of leaching, and contaminants removal during natural events of flash floods, the terraces and hill slopes are less likely to be exposed to significant natural wetting and flushing. Thus, the natural attenuation of the pollution in these terrains is expected to be significantly limited." (P. 2, L. 47-49).

Ln 74: The mounds are ponds. This I do not understand. How can a mound be a pond?

We are sorry for un-clarity of the text. The ponds are at the tops of the operational mounds. The chemical plant uses constructed ponds (on mounds tops) to allow the precipitation of the phosphogypsum from the acidic solution, thus gradually the bottom of the pond is filled and in order to preserve a pond, the phosphogypsum are mounted to the sides to form dikes around and elevate the pond. Thus gradually a mound of phosphogypsum Is formed. Text changed to be more clear (P3, L77).

Ln 141: Did you have field measurements of evaporation or measurements from which you could calculate potential evaporation?

As mentioned in the text: "field data of precipitation and evaporation, measured at a nearby meteorological station". While the precipitation data used for the simulation is the actual data during the period of measurements, evaporation data is the average daily potential evaporation in the region, as measured over a period of 9 years, as detailed in the manuscript (P. 3, L. 72).

Ln 148: There is no yellow star in figure 2B

The yellow star is in picture A and it shows the location where picture B was taken. Text was changed for clarification: "Yellow and red stars designate the location of the sandy unit (picture B) and the deep gorge unit (picture C), respectively."

Ln 172: If I understand the results that are shown correctly, the control experiments were conducted only once. Shouldn't the control experiments also have been repeated with two consecutive experiments?

Since the control had significantly lower initial concentrations for the different solutes, we didn't think there was a need to do a consecutive experiment. In the revised manuscript we clarified this point (P. 7 , L. 176-178).

Ln 205: Table 2, could you also include the monthly rain depth in the table?

Yes. Added. We also computed aridity index (P/PET) for each month.

Ln 229: You write that the extracted Na, K, and Mg likely originated from the dissolution of salts with a relatively high solubility. I can follow this argument, but could you give an estimate of how much of these cations could be adsorbed to the soil particles. What is the CEC of the soil?

Unfortunately, we do not have the ability to measure the soil CEC at this point. However, for the sake of simplicity, we prefer to focus the discussion on conservative tracers. This is also highlighted in the numerical model section.

Ln 250: The y-axis of figure 5 is confusing. It suggests that the ration of infiltration to runoff is shown. I suppose the ratios of infiltration to rain and of runoff to rain are shown.

Y-axis title and figure caption were changed.

Ln 260: I am wondering why you define the water:soil ratio based on the amount of water that was collected to the soil mass. I would define it based on the amount of water that was added at the time the water at the outlet was sampled. The water:soil ratio that you obtain is fully determined by the arbitrary amount of water that was collected before the first analysis.

We are not sure we completely understand this comment. The sampled water is the actual mass (or volume) of water that interacted with the soil up to the point of sampling. The sampled water passed through the soil, interacted with the minerals and solutes and transported the solutes out of the soil (to our sampling tube). This mass of water is what we analyzed, therefore we considered the mass of that sample in respect to the mass of soil it passed through.

Ln 345 Figure 7: Is it actual rain and actual evaporation or actual rain and potential evaporation? What is 'actual' rain?

Good point. Actual rain is measured rain. Title was changes to say: "Precipitation and PET". Figure caption was improved also to make sure there will be no confusion by the readers. Same change was done to Figure 9.

Ln 345: 'Precipitation and evaporation are presented in (I-J)' Shouldn't that be A and B?

Yes – fixed.

Ln 328-335: 'In the West plot however, the infiltrating water mobilized some of the Cl and Ca ions to depths greater than the lower boundary of the SEC. Beyond this depth the water and ions will not return to the soil surface by capillary flow. This may explain the 20% reduction in Ca concentration at the soil surface of the West plot in summer 2021 as compared to summer 2020 (**Fig.7H**). In contrast to the Ca concentrations, the Cl soil surface concentrations were similar for September 2020 and September. However, the center mass of the Cl contaminated zone was lowered from ~20 cm depth in summer 2020 to ~40 cm depth in summer 2021. A similar process may also take place with Ca, which may be obscured due to the higher Ca concentrations and the presence of phosphogypsum.' I am not following the line of reasoning here and I feel that there is an overinterpretation of the observations. If there would be more leaching of ions at the western plot and if the reduced accumulation of Ca at the soil surface at the end of the monitoring period could be explained by the fact that a substantial amount of Ca leached out the SEC, then the same should hold even more for the Cl because clearly more (in relative terms) Cl was below the SEC than Ca. Therefore, one would expect then a considerably lower accumulation of Cl at the end of the monitoring period but this was not observed.

We agree that this section was cumbersome. Following this comment and additional comment by Reviewer #1, the entire section was rewritten and simplified. Please see P.16, L.348-354.

Ln 375: try to improve the color scale since it is nearly impossible to see the zero-flux plane.

The scale was narrowed down.

In the discussion of the simulation results, I propose to show also the cumulative rainfall and cumulative simulated actual evaporation. I would expect that when the cumulative evaporation is smaller than the cumulative rainfall, that there must be a net downward movement of the solute. This net downward movement can be very small and then it is important to evaluate the movement and leaching over multiple years.

Good point. Please see reply below the last comment.

The question to my understanding is, whether and how the depth of the SEC can be used to evaluate whether there is a net downward movement of water (and solutes), which is the case when rainfall is larger than evaporation. The way the results are presented do not help me to answer these questions. My proposal would be to evaluate the fluxes simulated with Hydrus at the SEC and compare them with the assumption in the capacitor model about these fluxes. I presume that the assumptions in the capacitor model are that the fluxes are either close to zero or downward at the SEC. But, the overall cumulative flux at times when the fluxes are not downward should be very small compared to the cumulative evaporation at the surface. Doing this analysis for a set of 'hypothetical' SEC depths (other than the SEC depth that is estimated) could be used to evaluate the accuracy of the SEC depth estimation. I would assume that above the SEC, the cumulative upward fluxes are considerably larger than zero. The SEC depths would represent the minimal depths below which the cumulative upward fluxes are zero.

Good point. Please see reply below the last comment.

Conclusions Ln 380: I propose to include in the conclusion section the ratio of cumulative evaporation to rainfall. I suspect this ratio to be close to 1 for arid environments. Wouldn't this ratio explain the downward movement of the solutes? In fact, the movement of non-sorbing species is used to assess net groundwater recharge in very dry environments and there exists a lot of literature on this topic.

The last three comments are very good and helped to improve the paper considerably. Another analysis and figure (Figure 10) were added to the results section of the numerical model (Section 3.4). This analysis deals with the SEC water content, that is affected by net input of water into the soil. This analysis for the three examined precipitation conditions demonstrates the important impact that rain depth has on the potential of SEC drainage and solutes removal.

Following that, another section was added, entitled "*Applying the SEC model to predict the potential of solutes and contaminants leaching from different soil textures*". In this section the impact of soil textural properties, precipitation, and evaporation on Lc, SEC water content and drainage of the SEC are discussed and presented in a new figure (Figure 11). Lastly, these aspects were highlighted at the summary and conclusion section in P. 25, L485-490.

---

## Author Response (AR2)

Dear Editor,

We thank you and the reviewer for the constructive comments and thoughtful review. In the revised MS we addressed the major comments raised by you and the reviewer as well as the three specific comments provided by the reviewer, as detailed below.

As we understood the review report, major issues raised by the Editor and the Reviewer are:

- Ignoring cumulative precipitation vs. cumulative evaporation, and its impact on water leaching out of the SEC.
- SEC sensitivity to soil physical properties and accuracy of the V.G. Alfa parameter.

In line with the comments of the Reviewer and the Editor, section 3.5 was modified to include an analysis of the impact of cumulative evaporation and precipitation on SEC thickness and potential water flow out of the SEC. Following Lehman et al. (2019) the analysis was done for monthly time intervals. However, in contrast to Lehman et al., which neglected the cumulative evaporation, our analysis includes cumulative values of both monthly evaporation and precipitation. For each month we computed expected drainage out of the SEC by the cumulative evaporation-precipitation analysis (Equation 10 in the revised MS), as well as predicted leaching values as computed by the HYDRUS simulation, for the three examined climatic scenarios. We believe this new addition further demonstrates the validity of the SEC concept in estimating water flow and consequent solute transport processes near the soil surface. The added section is in P25-26, L489-526 and includes Figure 13.

The Reviewer suggested to focus on the arid conditions and to ignore the cumulative impact. This is in contradiction to the Editor's comment as detailed above. In the revised MS we discuss the validity of the SEC model and needed assumptions for arid conditions and wetter conditions. We emphasized that Figure 12 is relevant for arid conditions, whereas the new figure (figure 13) and the cumulative precipitation/evaporation approach is more relevant to wetter conditions. We hope that by doing so we managed to address both comments.

Lastly, the reviewer was concerned about the sensitivity of the SEC model to soil physical properties, namely the Alfa parameter. Upon the reviewer request we added Figure 4 that presents the water retention curves of three soil samples. In accordance to Figure 4 we discuss the observed variations in V.G. parameters and their impact on the SEC (P10-11, L237-250, and P23, L 473-479). Moreover, Figure 11 (Figure 12 in the revised MS) includes now more data points that includes the different soil samples from the field and the modelled soil that was derived from these measurements.

**Specific comments:**

*C1: Lines 196 and 199: don't use 'n' for porosity .... (line 112, van Genuchten n)`*
>    R1: Accepted, changed to $\emptyset$.

*C2: Line 358: why is the critical water content changing when it depends only on parameter m (equation 5)?*
>    R2: This is true. Our mistake, Changed.

*C3: Table 3: bulk density unit is g/cm3*
>    R3: Fixed.

Following these changes, we hope you will find the paper suitable for publication.

With best regards.

---

## Author Response (AR3)

Dear Editor,

We sincerely appreciate your decision to accept our manuscript, "Soil contamination in arid environments and assessment of remediation applying surface evaporation capacitor model; a case study from the Judean Desert, Israel," for publication in SOIL, pending minor technical corrections.

As requested, we have made the following revisions:

We have narrowed the color scales of subplots C, E, and G in Figure 10.

We have added sections detailing "Competing Interests," "Author Contributions," and "Financial Support".

We believe these revisions address your feedback and that the manuscript is now suitable for publication in its current state.

Thank you for your consideration.

Sincerely,

Rotem Golan and Uri Nachshon.